# A new extended belief rule base method based on neighborhood covering reduction for diabetes diagnosis

Shucheng Feng[1], Wei He🔘[1]*, Li Jiang[2]*, Manlin Chen[3]

1 Harbin Normal University, Harbin, China, 2 Harbin Medical University Cancer Hospital, Harbin, China,
3 ChangChun University of Technology, Changchun, China

* hewei@hrbnu.edu.cn, 601935@hrbmu.edu.cn

## Abstract

The precise diagnosis and scientific management of diabetes are highly important for improving patients' quality of life and reducing the risk of complications. However, in actual clinical settings, diagnostic processes often face challenges, including significant individual differences among patients, complex and diverse parameters, and heterogeneity in disease progression. These challenges not only impose greater requirements on the adaptability and precision of diagnostic and therapeutic models but also highlight the need for explainable disease mechanisms and rational treatment strategies. To address these issues, this study proposes an Extended Belief Rule Base (EBRB) model based on neighborhood covering reduction, abbreviated as NCR-EBRB, for diabetes prediction and diagnosis. During the model construction phase, the Extreme Gradient Boosting (XGBoost) method is first employed for feature importance evaluation to reasonably screen key features and effectively reduce model dimensionality. In the model inference phase, the Neighborhood Covering Reduction (NCR) method is adopted to implement rule reduction in the rule base, combined with a threshold-based rule activation strategy to filter out inefficient rules, ensuring efficient reasoning processes and effective result output. During the model optimization phase, the Projection Covariance Matrix Adaptive Evolution Strategy (P-CMA-ES) is applied to optimize the parameters of the streamlined rule base, aiming to identify optimal parameter configurations for further improving model performance. Through this meticulous parameter tuning, the diagnostic accuracy is enhanced, and the robustness of the model is improved.

## 1. Introduction

Diabetes is characterized as a chronic metabolic disorder characterized by hyperglycemia [1], affecting the health of hundreds of millions of people worldwide. If not properly controlled, it may lead to various severe complications, including cardiovascular

**Data availability statement:** All relevant data are within the paper and its Supporting Information files. The original diabetes data is available from Mendeley (https://data.mendeley.com/datasets/wj9rwkp9c2/1) and the validation datasets from UCI (https://archive.ics.uci.edu/). The 'Supporting Information' ZIP file includes both the original data (in the 'origin' folder) and the specific processed splits (two-attribute version) used to generate the results.".

**Funding:** This work was supported in part by Open Foundation of Key Laboratory of the Ministry of Education on Application of Artificial Intelligence in Equipment under Grant No.AAIE-2023-0103, in part by the Natural Science Foundation of Heilongjiang Province under Grant No. PL2024G009, in part by the Basic Research Support Program for Outstanding Young Teachers in Provincial Undergraduate Universities of Heilongjiang Province under Grant No. YQJH2024116.

**Competing interests:** The authors have declared that no competing interests exist.

diseases [2], kidney failure [3], retinopathy [4] and foot damage [5]. Therefore, early prevention of diabetes and rational regulation of blood glucose levels are crucial for reducing the incidence of complications, improving patients' quality of life, and alleviating societal healthcare burdens.

Extensive research has been conducted on diabetes prediction. Dinh A [6] adopted supervised machine learning methods, including logistic regression, support vector machines, random forests, and gradient boosting, to construct models for diagnosing cardiovascular diseases, prediabetes, and diabetes. Multiple algorithms were evaluated across different feature sets and time periods. A weighted ensemble model was developed to increase diagnostic accuracy, with tree-based models being utilized to identify key variables in disease diagnosis. Nadeem M W [7] proposed a multi-layer machine learning fusion-based decision support system for diabetes identification. The system architecture comprises data sources, data fusion, and machine learning fusion layers. It integrates multiple data streams, including electronic health records, with K-fold cross-validation being employed for hyperparameter selection. Support Vector Machine (SVM) and Artificial Neural Network (ANN) classifiers were fused through posterior probability methods to improve diabetes classification accuracy. Yeliz Kaya [8] developed a gestational diabetes prediction system based on multiple machine learning algorithms. The study integrated Extra Trees Classifier, Voting Classifier, Light Gradient Boosting Machine Classifier, Extreme Gradient Boosting Classifier, Logistic Regression, and Random Forest Classifier. Through analysis of pregnant population data, the Extreme Gradient Boosting Classifier was found to demonstrate optimal performance across two independent prediction models, with its predictive effectiveness being significantly superior to that of other comparative methods. Qin Y F, [9] developed multiple machine learning models for diabetes prediction based on lifestyle data from large-scale health databases, with key predictive variables being screened through feature selection methods. The study revealed that the Gradient Boosting algorithm exhibited outstanding performance in this prediction task, achieving not only high classification accuracy but also good predictive efficacy through area under the Receiver Operating Characteristic Curve (AUC-ROC) metrics. This research specifically highlighted the significant role of dietary energy and macronutrient intake levels in diabetes prediction, emphasizing the technical advantages of machine learning in lifestyle-related disease risk assessment. Bukhari M M, [10] developed an improved Artificial Neural Network (ANN) diabetes prediction model for large-scale health databases. The model employed Artificial Backpropagation Scaled Conjugate Gradient Neural Network (ABP-SCGNN) algorithms and outperformed other tested neural network models, including General Regression Neural Network (GRNN), Multilayer Perceptron (MLP) Neural Network, Radial Basis Function (RBF) Network, and Feedforward Neural Networks with various backpropagation training algorithms.

On the basis of the aforementioned research achievements, diabetes prediction models still have room for improvement. Existing models are predominantly based on data-driven black-box approaches, which, while achieving high accuracy, lack interpretability. This limitation has restricted experts' understanding of disease etiology

[11]. To address this issue, a grey-box model driven by expert knowledge and data is proposed in this study, with the aim of enhancing the model's explanatory ability.

The EBRB [12] serves as a dual-driven model that integrates knowledge and data and embeds expert knowledge into data-driven models by extending antecedent attributes into belief distributions, thereby enabling better representation of knowledge uncertainty. It has been successfully applied in sensor-based activity recognition [13], environmental governance cost prediction [14], environmental investment forecasting [15], carbon emission prediction [16], thyroid nodule diagnosis [17], and other fields.

However, since the EBRB is constructed on a data-driven rule basis, the selection of activation rules from massive data samples has become the primary research challenge. Yu R [18] introduced the 80/20 rule, which activates only the top 20% most significant rules for given inputs. Su Q [19] proposed a framework based on Burkhard-Keller (BK) trees to optimize the structure of the EBRB system. BK trees are used to index rules and retrieve those close to input data through search threshold settings, with nearest-neighbor rules being activated for final Evidential Reasoning (ER) synthesis. Lin Y Q. [20] developed an EBRB system using Vantage Point (VP) trees and Multi-Vantage Point (MVP) trees, where k-means clustering is employed to select appropriate query thresholds. Nevertheless, determining the optimal number of clusters remains challenging. Lin Y Q [21]. integrated BK-trees and KD-trees to develop a multi-attribute search framework for reconstructing relationships within EBRB systems. This method relies primarily on data dimensionality to switch between two tree structures. The experimental results demonstrate that this approach achieves high accuracy and efficient retrieval performance.

The scale of rules is found to affect reasoning efficiency and final outcomes. Therefore, the efficiency of EBRB systems can be fundamentally enhanced through rule deletion approaches. In light of this, an EBRB model based on Neighborhood Covering Reduction (NCR-EBRB) is proposed in this study.

The main contributions of this study are outlined as follows:

1. An EBRB method is proposed for diabetes diagnosis, providing a novel approach to diabetic disease assessment.

2. To address the issue of excessive rule quantities in EBRB systems, a neighborhood covering reduction algorithm is introduced. This algorithm is implemented to conduct neighborhood analysis for EBRB systems, where concise rule representations are used to replace redundant rule bases, thereby enhancing reasoning efficiency.

The remainder of this study is organized as follows. Section 2 clarifies three critical challenges that must be addressed when constructing EBRB models for diabetes diagnosis. Section 3 details the primary procedures for building the NCR-EBRB framework. Section 4 presents diagnostic case studies and benchmark experiments on diabetes diagnosis to validate the effectiveness of the proposed model. Finally, Section 5 concludes the research.

## 2. Problem description

When a diabetes diagnostic model is constructed using the EBRB, three problems need to be solved:

Problem 1: During the establishment of the original EBRB model, how the antecedent attributes of the EBRB can be effectively defined. As a dual-driven hybrid model that integrates data and expert knowledge, the EBRB possesses inherent interpretability. However, excessive features may lead to model complexity that hinders the interpretation of individual feature impacts on final decisions, thereby diminishing model explainability. Furthermore, not all features necessarily contribute to prediction accuracy in most scenarios. The introduction of irrelevant features may introduce additional noise to the model, potentially reducing diagnostic precision. This can be described as:

$$\mathbf{A} = \{a_1, \ldots, a_M\} \rightarrow \mathbf{A}' = \{a_1, \ldots, a_N\} \, N \le M \tag{1}$$

where $\mathbf{A}$ denotes the original attribute set, and $\mathbf{A}'$ represents the filtered attribute set. $M$ and $N$ indicate the number of attributes in the original and filtered attribute sets, respectively, and $a_i(i \le M)$ represents the $i$th attribute.

Problem 2: How to avoid low reasoning efficiency caused by excessive rule quantities in the rule base. In the EBRB model, rules are generated by converting antecedent attributes into belief distributions. When the training set contains relatively few samples, this one-to-one correspondence does not cause significant issues. However, as the training set scale increases, the rule base undergoes rapid expansion, leading to severe degradation of model reasoning efficiency. On the one hand, this rule-based expansion increases the computational complexity and storage demands; on the other hand, oversized rule bases may introduce redundant information, thereby reducing the model's generalization capability and overall performance. Therefore, to improve the application effectiveness of EBRB models, effective strategies must be developed to control and optimize the rule-based scale, aiming to preserve model advantages while enhancing reasoning efficiency and practicality. This can be formally described as follows:

$$\Theta = \{R_1, R_2, \cdots, R_j\} \rightarrow \Theta' = \{R_1, R_2, \cdots, Rk\}\, k \leq j \tag{2}$$

where $\Theta$ is denoted as the original EBRB rule base, $\Theta'$ represents the fused rule set after applying a specific method, $j$ indicates the total number of rules in the original rule base, $k$ represents the total number of rules in the optimized rule base, $Ri(i \leq k)$ corresponds to the $i$th rule, and it is evident that the number of rules in $\Theta'$ is smaller than that in $\Theta$.

Problem 3: How to further improve model accuracy. In the original EBRB system, parameters such as rule weights and attribute weights are assigned primarily on the basis of expert knowledge. However, with the continuous expansion of dataset scales, it has become increasingly difficult for experts to allocate weights precisely for each rule and attribute, which may lead to significant degradation in model accuracy and effectiveness. Consequently, how to scientifically and efficiently determine these weights in large-scale data environments has emerged as a critical issue requiring urgent resolution for enhancing EBRB model performance. Automated or semi-automated weight allocation methods should be developed, which would not only alleviate expert workloads but also better capture implicit patterns within data, thereby improving the model's overall performance and adaptability. This can be described as:

$$P = (x, y, \gamma) \rightarrow P_{best} = \text{optimize}\, (x, y, \gamma') \tag{3}$$

where: P is defined as the original parameter set, $P_{best}$ is defined as the optimized parameter set, $x$ denotes the dataset, $y$ represents the evaluation result, $\gamma$ corresponds to the original parameter, $\gamma'$ indicates the optimized parameter, and optimize $(.)$ is the optimization function.

## 3. Diabetes diagnosis model based on the NCR-EBRB

This study proposes a novel NCR-EBRB model based on Neighborhood Covering Reduction (NCR), whose construction involves three sequential steps: 1. Model Construction Phase: Noncritical features are filtered via XGBoost technology to construct the initial model. 2. Model inference phase: Rule reduction is first implemented through neighborhood coverage analysis, followed by threshold-based activation constraints to limit the number of active rules during inference. Model Optimization Phase: Parameter training is conducted via the projection covariance matrix adaptation evolution strategy (P-CMA-ES) algorithm to further optimize the proposed NCR-EBRB system.

For a comprehensive understanding, the framework of the proposed model is illustrated in Fig 1.

### 3.1. Model construction

The model construction of the NCR-EBRB is divided into two steps: 1. The XGBoost technique is employed to evaluate feature importance [22], and non-important features are filtered out. The filtered attributes are defined as antecedent attributes, and the rule base is constructed based on the EBRB rule base construction process.

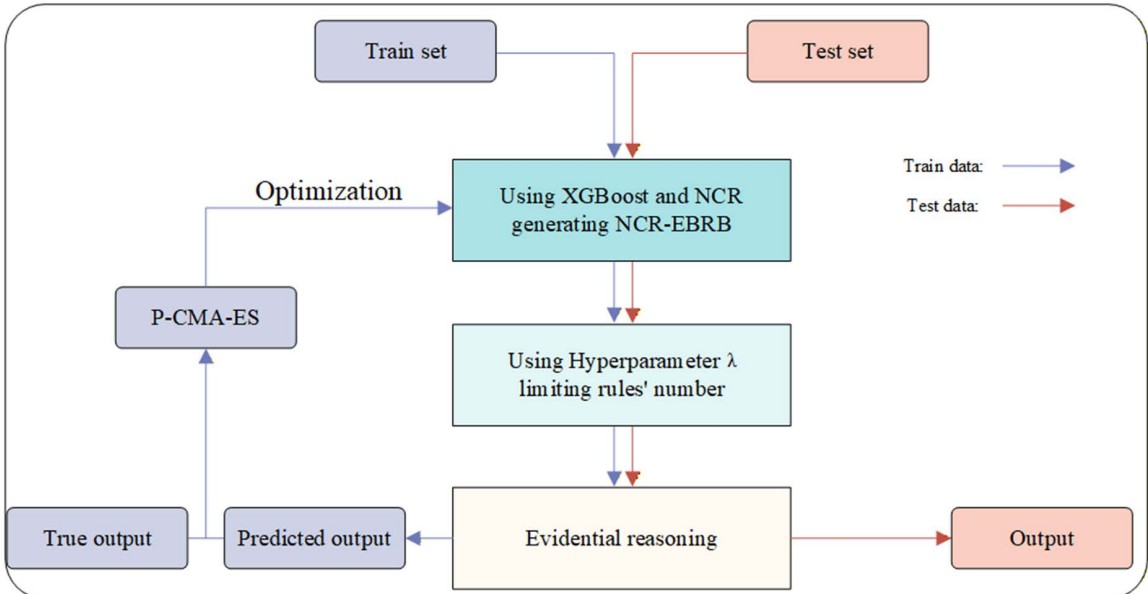

**Fig 1. Overall structure of the model.**

**3.1.1. XGBoost feature selection.** Prior to model construction, the XGBoost method is first utilized to define the antecedent attributes. The specific procedure is outlined as follows:

A dataset containing $n$ samples and $M$ feature values is defined as:

$$\hat{y}_i = \sum_{t=1}^{T} f_t(x_i), f_t \in F$$

(4)

where $\hat{y}_i$ is the predicted value, $t$ represents the $t$th decision tree, $T$ denotes the total number of decision trees, $f_t$ indicates the structure of a decision tree, and $F$ represents the tree space. The objective is to minimize Equation (5).

$$Obj = \sum_{i=1}^{n} L(y_i, \hat{y}) + \sum_{i=1}^{t} \Omega(f_i)$$

(5)

where $Obj$ is the optimization objective, $L(g)$ is the loss function, and $\Omega(f_i)$ is the regularization term. The loss function and gain function are given by Equations (6) and (7), respectively.

$$L^{(t)} = \sum_{i=1}^{k} \left[ l\left(y_i, y_i^{t-1} + g_i f_t(x_i) + \frac{1}{2} h_i f_t^2(x)\right) \right] + \Omega(f_t)$$

(6)

In the equation, $L$ represents the loss function, $L^{(t)}$ denotes the $t$th iteration of $L$, $\Omega(f_i)$ is the regularization term, which represents the second-order Taylor series of $L$ at the $t$th iteration, and $g_i$ and $h_i$ indicate the first-order and second-order

gradients, respectively. In this study, the gain function described in the equation uses Equation (7) as the metric for determining the optimal split node.

$$gain = \frac{1}{2}\left[\frac{\left(\sum\limits_{\in I_L} g_i\right)^2}{\sum\limits_{\in I_L} h_i + \eta} + \frac{\left(\sum\limits_{\in I_R} g_i\right)^2}{\sum\limits_{\in I_R} h_i + \eta} - \frac{\left(\sum\limits_{\in I} g_i\right)^2}{\sum\limits_{\in I} h_i + \eta}\right] - \gamma \tag{7}$$

In equation, $gain$ represents the gain function; $I_L$ and $I_R$ denote the samples in the left and right nodes after splitting; and $I = I_L \bigcup I_R$, $\eta$ and $\gamma$ are penalty parameters. The gain indicates the gain score of each split in the tree. The final feature importance score is calculated by the average gain. The average gain is defined as the total gain across all trees divided by the total number of splits for each feature. A higher feature importance score implies greater significance of the corresponding feature.

In the EBRB rule-based construction process of this study, the two most important parameters are selected for building the rule base.

**3.1.2. Establishing the EBRB rule base.** As an expert system, the BRB is composed of a series of belief rules (BRs). The EBRB, as a variant of the BRB, is similarly constructed from multiple extended belief rules (EBRs). The extended belief rule (EBRB) can be expressed as:

$$R_k : \text{IF} U_1 \text{is} \left\{\left(A_{1,1}, \alpha^k_{1,1}\right), \left(A_{1,2}, \alpha^k_{1,2}\right), \cdots, \left(A_{1,J_1}, \alpha^k_{1,J_1}\right)\right\} \wedge \cdots$$
$$\wedge U_M \text{is} \left\{\left(A_{M,1}, \alpha^k_{M,1}\right), \left(A_{M,2}, \alpha^k_{M,2}\right), \cdots, \left(A_{M,J_M}, \alpha^k_{M,J_M}\right)\right\}$$
$$\text{THEN} \left\{\left(D_1, \beta^k_1\right), \left(D_2, \beta^k_2\right), \cdots, \left(D_N, \beta^k_N\right)\right\}$$
$$\text{With a rule weight } \theta_k \text{ and attribute weights } \left\{\delta_1, \delta_2, , \delta_M\right\}.$$
$$\text{s.t.} 0 \le \alpha^k_{ij}, \beta^k_j, \delta_i, \theta_k \le 1,$$
$$\sum_{j=1}^N \beta^k_j \le 1, \sum_{j=1}^{I_i} \alpha^k_{ij} \le 1, i \in \{1, \cdots, M\} \tag{8}$$

where $k$ is the rule index. $U_i$ $(i = 1, 2,..., M)$ represents the $i$th antecedent attribute, $M$ denotes the number of antecedent attributes, $A_{i,j}$ $(j = 1, 2,..., J_i)$ is the $j$th reference value of the $i$th antecedent attribute, $J_i$ is the number of reference values, $\alpha^k_{i,j}$ $(k = 1, 2,..., L)$ represents the matching degree of the corresponding reference value, $L$ indicates the total number of rules in the rule base, $D_n$ (n = 1, 2, . . . , N) denotes the $n$th reference value of the consequent attribute, and $\beta^k_n$ represents the belief degree of the $n$th reference value in the consequent of the $k$th rule (if $\sum_{n=1}^N \beta^k_n = 1$, the rule is complete; otherwise, it is incomplete). The hyperparameters in the model are $\delta_i$ and $\theta_k$, where $\delta_i$ is the attribute weight, which represents the importance of the $i$th attribute, and $\theta_k$ $(k = 1, \cdots, L)$ is the rule weight.

The construction of the EBRB system differs from that of the BRB system in that the EBRB system is generated on the basis of existing data, with specific generation rules as follows:

Step 1: On the basis of expert knowledge, the reference values $A_{i,j}$ for each antecedent attribute and the $D_n$ consequent attributes are determined.

Step 2: Using the $A_{i,j}$ and $D_n$ generated in Step 1, the input data $x^k = \left(x^k_1, x^k_2, \cdots, x^k_M\right)$ are transformed into a corresponding belief distribution of the following form:

$$E\left(x^k_i\right) = \left\{\left(A_{i,j}, \alpha^k_{i,j}\right), j = 1, 2, \cdots, J_i\right\}, i = 1, \cdots, M \tag{9}$$

Let $u(A_{i,j})$ represent the utility value of the reference value $A_{i,j}$; simultaneously, it is necessary to ensure that $u(A_{i,j}) < u(A_{i,j+1})$ $(j = 1, 2, ...J_i - 1)$. Then, $\alpha_{i,j}^k$ is calculated as follows:

$$\begin{aligned}
\alpha_{i,j}^k &= \frac{u(A_{i,j+1}) - x_i^k}{u(A_{i,j+1}) - u(A_{i,j})}, u(A_{i,j}) \leq x_i^k \leq u(A_{i,j}) \\
\alpha_{i,j+1}^k &= 1 - \alpha_{i,j}^k, \quad u(A_{i,j}) \leq x_i^k \leq u(A_{i,j}) \\
\alpha_{i,t}^k &= 0, \quad t = 1, \cdots, j-1, j+2, \cdots, J_i
\end{aligned}$$

(10)

For the consequent component $y^k$ corresponding to $x^k$, the subsequent form of the belief distribution for evaluation levels can also be calculated via a method similar to Equation (9).

$$E(y^k) = \{(D_n, \beta_n^k), n = 1, 2, \cdots, N\}$$

(11)

Through the aforementioned calculations, each piece of data is transformed into a rule form with belief distributions, thereby constructing the initial rule base.

Step 3: Determine the parameters of the EBRB system, including the attribute weights and rule weights. In the original EBRB, the rule weights and attribute weights are provided by experts.

Fig 2 illustrates the rule construction process of the NCR-EBRB model.

## 3.2. Model inference phase

During the inference stage, on the one hand, the hierarchical reduction of the rule base is achieved through adaptive adjustment of the neighborhood coverage reduction; on the other hand, the evidential reasoning engine is employed to fuse the activated rules.

**3.2.1. Neighborhood covering reduction.** First, we briefly introduce the preliminaries of neighborhood coverage reduction. [23–25]

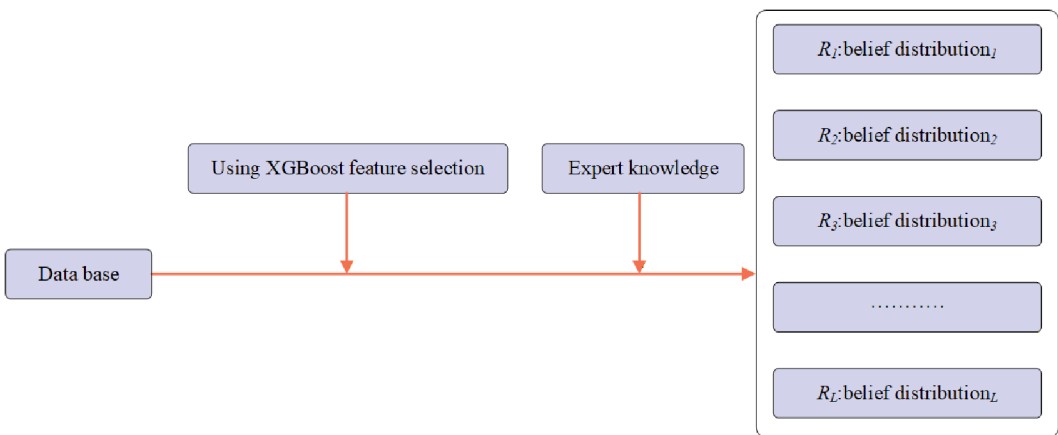

**Fig 2. Construction process of the NCR-EBRB rules.**

Define 1. Neighborhood covering

Let $U$ be the data space of sample $\{x_1, x_2, ...x_n\}$, and $\delta(x)$ be the neighborhood of $x$, which is defined by the formula $O(x) = \{x_i \Delta(x, x_i) \leq \eta\}$, where $\eta$ is a threshold and $\Delta$ is a distance function representing the distance between any two sample points. The complete set of all neighborhoods is denoted as $O = \{O(x_1), O(x_2), ..., O(x_n)\}$, and the union of these neighborhoods $O = \bigcup_{i=1}^{n} O_i$ forms coverage of the data space. The neighborhood coverage is denoted as $C = \langle U, O \rangle$.

Specifically, for each sample $\{x_1, x_2, ...x_n\}$ in the data space, its neighborhood $\delta(x)$ contains all points whose distance from $\{x_1, x_2, ...x_n\}$ does not exceed the threshold $\eta$. The set of neighborhoods defined in this way can cover the entire data space, meaning that all the data points belong to at least one neighborhood. This coverage method is termed neighborhood coverage.

In data coverage, neighborhoods frequently overlap, and some neighborhoods may be redundant in preserving the data distribution structure. To reveal the fundamental structure of the data space, eliminating these redundant neighborhoods is critical, thereby obtaining more compact and efficient coverage.

By removing neighborhoods that are unnecessary for representing the data distribution structure, the coverage can not only be simplified but also enhance the efficiency of data analysis and processing. A more streamlined coverage helps reduce computational complexity and provides a clearer representation of the key features and patterns inherent in the data.

Define 2. Neighborhood covering reduction

Let $C = \langle U, O \rangle$ be a coverage of the data space. For a neighborhood $O(x) \in O$, after removing $O(x)$ from the neighborhood set, a new set $O' = O - \{O(x)\}$ is obtained. If the union of the remaining set still covers the entire data space, i.e., $\bigcup_{O(x_i) \in O} O(x_i) = \bigcup_{O(x_j) \in O'} O(x_j)$, then the neighborhood $O(x)$ is considered reducible; otherwise, it is irreducible.

Furthermore, if every element $O$ in the neighborhood set $C = \langle U, O \rangle$ is irreducible, the coverage $C$ is termed irreducible; otherwise, $C$ is termed reducible.

Definition 3. Relative neighborhood covering reduction

Let $C = \langle U, O \rangle$ be a neighborhood coverage, $X \subseteq U$ be the sample set, and $O(x_i) \in O$ be a neighborhood. If $\exists O(x_j) \in O$ such that $O(x_i) \subseteq O(x_j) \subseteq X$, then $O(x_i)$ is considered a relatively reducible neighborhood with respect to $X$; otherwise, it is relatively irreducible. If all neighborhoods in $C$ are relatively irreducible, the neighborhood coverage $C$ is termed relatively irreducible.

Define 4. Measure the Sample Similarity

To establish neighborhoods within the dataset, the distances between samples are first calculated. Given that each rule in the EBRB rule base consists of numerical data, the Euclidean distance is adopted to calculate the similarity between rules.

Define 5. Reduction

Given a sample $x \in U$„ its neighborhood $O(x)$ is composed of data points close to $x$.

$$O(x) = \{y \mid \Delta(x, y) \leq \eta, y \in U\} \tag{12}$$

where $\Delta$ is the distance function and where $\eta$ is the distance threshold, which represents the radius of the neighborhood. When the neighborhood $O(x)$ of a sample $x$ is computed, all data points whose distance from $x$ is less than or equal to $\eta$ are included within this neighborhood.

Relying solely on threshold-based neighborhood radius settings may introduce certain limitations. Therefore, the concepts of nearest hit $NH(x)$ and nearest miss $NM(x)$ are further introduced to enhance the granularity of analysis. For a

sample $x$, its nearest hit is defined as the closest sample belonging to the same class. If a class contains only one sample, the sample itself is designated its nearest hit. Conversely, the nearest miss is defined as the closest sample belonging to a different class from $x$.

On the basis of the above definitions, the classification margin is defined as the difference between the distances from a sample $x$ to its nearest miss and nearest hit. Specifically, for a given data point $x$, two critical neighbors are defined: the nearest hit $NH(x)$ and the nearest miss $NM(x)$. The nearest hit is the closest instance sharing the same class as $x$, whereas the nearest miss is the closest instance belonging to a different class. If a class contains only one member, the nearest hit for that member is designated as itself. Ultimately, the classification margin of the data point $x$ is determined by the difference in its distances to these two critical neighbors. The formula for calculating the classification margin is as follows:

$$m(x) = \Delta(x, NM(x)) - \Delta(x, NH(x))$$

(13)

Notably, the classification margin $m(x)$ can potentially take negative values. These cases indicate that the sample may be misclassified when the nearest neighbor rule is used. To address this issue, any negative classification margin is adjusted to zero, i.e., $m(x) = 0$ is set in such cases. Fig 3 illustrates the reduction results of a dataset after nearest neighbor analysis.

NCR ensures reasoning completeness by maintaining rule coverage sufficiency (i.e., all samples are covered by at least one rule). Meanwhile, NCR preserves semantic integrity by selecting rules without altering their structures (including semantic attributes and belief distributions) and prioritizes expert-defined critical decision boundary rules via classification margin analysis (Eq. 13).

**3.2.2. Search the rule base.** The previous sections introduced the core relative neighborhood covering component in the rule reduction algorithm. However, generating a small set of rules that cover the original rules remains a challenging problem, as the search for a minimal rule set is an NP-hard problem [26]. Nevertheless, multiple strategies exist for searching suboptimal rule sets, such as forward search, backward search, and genetic algorithms [27]. In this study, the forward search technique is considered, which begins with an empty rule set and incrementally adds new rules one by one. At each step, the consistent neighborhood that covers the most samples is selected to generate a rule, achieving $O(Rn^2)$ time complexity. The specific steps are as follows:

**Algorithm 1: EBRB rule reduction algorithm based on neighborhood covering**

```
Input: Original rule base R = {R₁, ..., Rn}
Output: rule set R_new
1: compute the margins of rules m(Ri), i = 1, 2, ···n, if m(Ri) < 0, set m(Ri) = 0.
2: compute O(Ri) of sample Ri, i = 1, 2, ···n
3: O ← {O(Ri), i = 1, 2, ···n}, R_new ← ∅
4: compute the number of the samples covered by each covering element in O
5: While (O ≠ ∅)
6: select the covering element O(Ri) covering the largest number of samples
7: add a rule Ri into R_new
8: remove O(Rj) if O(Rj) ⊆ O(Ri), i ≠ j
9: end
```

In the forward search step, the algorithm greedily searches for the maximum neighborhood of samples, and rules covering the fewest samples are removed. Through this approach, a small set of rules capable of covering the majority of the original rules is generated for the rule base.

**3.2.3. Reasoning of the EBRB.** Once the EBRB rule base is constructed, reasoning can be performed on the query data to obtain outputs.

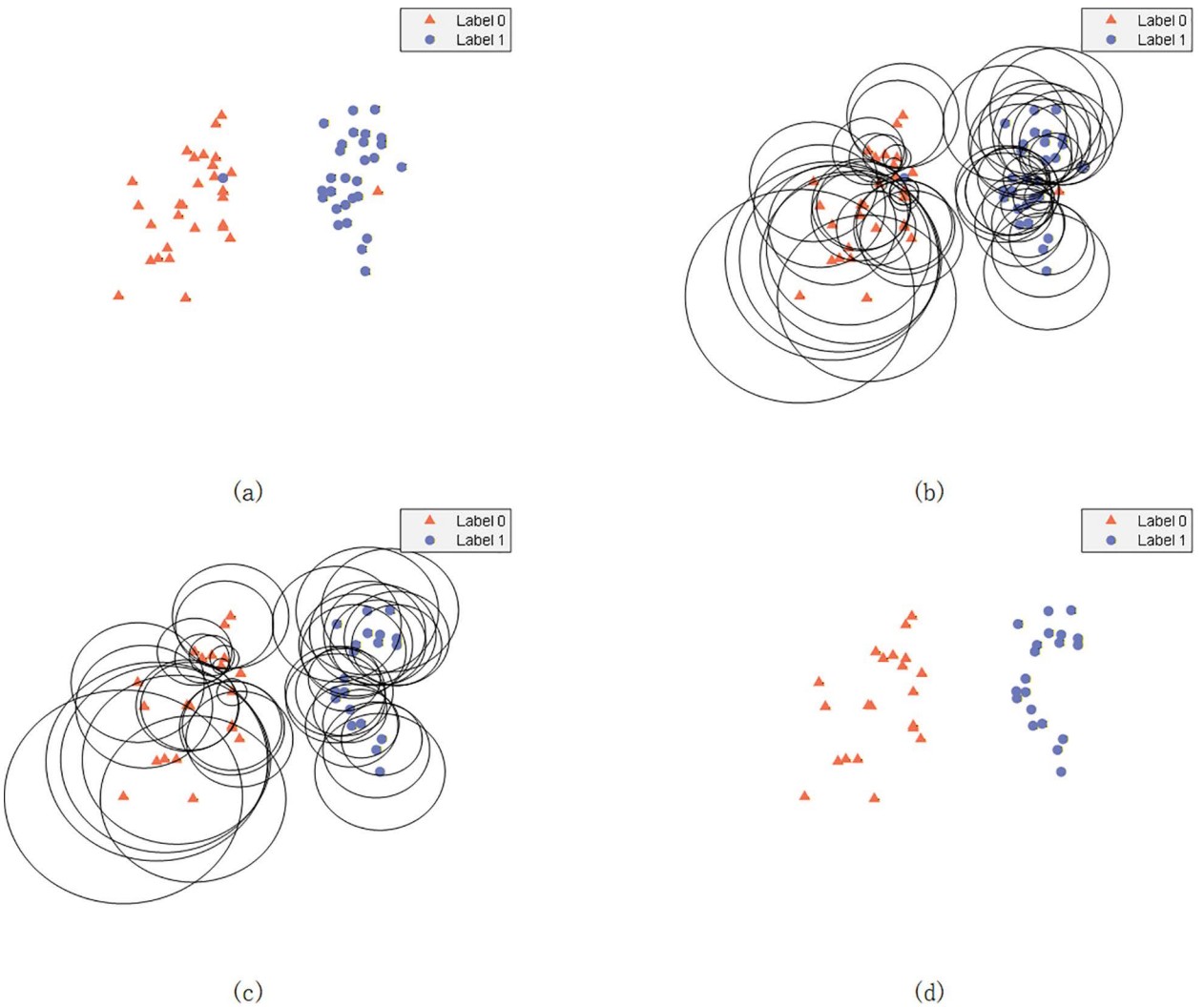

**Fig 3. Neighborhood coverage reduction for noisy data: (a) original data, (b) calculation of the neighborhood coverage, (c) neighborhood coverage reduction, and (d) result.**

Step 1: The query data are converted into the form of belief distributions, as described in Section 2.1. This ensures that the query data maintain consistency in representation format with the data in the existing rule base, thereby facilitating subsequent reasoning and analysis.

Step 2: The activation weight $\omega_k$ for each rule in the rule base due to data input is calculated, as all the rules in the rule base are activated upon data input. The distance between the input data and the rules is computed via the Euclidean distance method, as shown below:

$$S_i^k = 1 - d_i^k = 1 - \sqrt{\sum_{j=1}^{J_i} \left( \alpha_{ij} - \alpha_{ij}^k \right)^2}$$

$$\omega_k = \frac{\theta_k \times \prod_{i=1}^{M} \left( S_i^k \right)^{\bar{\delta}_i}}{\sum_{l=1}^{L} \left[ \theta_l \times \prod_{i=1}^{M} \left( S_i^k \right)^{\bar{\delta}_i} \right]}, \bar{\delta}_i = \frac{\delta_i}{\max\limits_{t=1,2,\cdots,M} \{ \delta_t \}}$$

(14)

Here, $0 \leq \omega_k \leq 1$ $(K = 1, \cdots, L)$, $\sum_{i=1}^{L} \omega_k = 1$; if $\omega_k = 0$, this implies that the $k$th rule is not activated.

Step 3: For rules satisfying $0 \leq \omega_k \leq 1$ the evidential reasoning algorithm is employed to fuse them, thereby generating the inference result $S(x)$.

$$S(x) = (D_j, \beta_j), j = 1, 2, \cdots, N \tag{15}$$

where

$$\beta_n = \frac{\mu \times \prod_{k=1}^{L} \left( \omega_k \beta_n^k + 1 - \omega_k \sum_{i=1}^{N} \beta_i^k \right) - \prod_{k=1}^{L} \left( 1 - \omega_k \sum_{i=1}^{N} \beta_i^k \right)}{1 - \mu \times \prod_{k=1}^{L} (1 - \omega_k)} \tag{16}$$

$$\mu = \left[ \sum_{j=1}^{N} \prod_{k=1}^{L} \left( \omega_k \beta_j^k + 1 - \omega_k \sum_{i=1}^{N} \beta_i^k \right) - (N-1) \prod_{k=1}^{L} \left( 1 - \omega_k \sum_{i=1}^{N} \beta_i^k \right) \right]^{-1} \tag{17}$$

For classification problems, the final result is determined via the following approach:

$$f(x) = D_j, j = \arg \max_{n=1,\ldots,N} \hat{\beta}_n \tag{18}$$

### 3.3. Model optimization phase

Model optimization can be divided into two aspects: the optimization of the reasoning engine and the optimization of relevant parameters.

**3.3.1. Optimization of the evidence reasoning (ER) approach.** Prior to parameter optimization, the original ER reasoning engine must be optimized. This is because although the NCR-EBRB rule base established in Section 3.2 can effectively replace the original redundant rule base with fewer rules, this does not guarantee the absence of redundant rules in the constructed rule base. Therefore, further optimization of the reasoning engine is needed.

In traditional EBRB systems, when the rule base is established and the input data to be evaluated are provided, the rules in the rule base are activated. For rules satisfying $0 \leq \omega_k \leq 1$ $(K = 1, \cdots, L)$, the ER reasoning engine is employed for inference. However, during the rule fusion process in classification EBRB systems, the following issue arises: a rule with a high matching degree and numerous redundant rules with extremely low matching degrees are simultaneously activated and fused. Intuitively, the inference result should align with the rule of a high degree of matching. However, the presence of many redundant rules with low matching degrees leads to inaccuracies in the final inference. Therefore, a method must be adopted to restrict the number of activated rules. In this study, a threshold $\lambda$ is applied to limit the number of rules. $\lambda$ can represent either the specific quantity of rules to be fused or the percentage of rules to be fused relative to the total activated rules.

**3.3.2. Optimization of NCR-EBRB parameters via P-CMA-ES.** In traditional EBRB models, rule weights $\theta_k$, attribute weights $\delta_i$ and consequent attributes $\beta_n^k$ are typically manually assigned by experts. However, in practical applications, as the number of rules increases, it becomes challenging for experts to provide precise values for each parameter. Therefore, the introduction of a parameter training mechanism becomes particularly critical. Through iterative optimization of the learning directions, more accurate information can be effectively aggregated, thereby significantly enhancing the performance of the NCR-EBRB model.

Since the model proposed in this study is applied to classification tasks, the consequent attribute $\beta_n^k$ is determined by the category to which the samples belong; thus, parameter optimization is not performed on the consequent attribute $\beta_n^k$.

$$P = (\theta_k, \delta_i) \tag{19}$$

where $P$ denotes the parameters of the NCR-EBRB. The parameter optimization model of the NCR-EBRB system is as follows:

$$\begin{aligned} \min &Loss\,(\theta_k, \delta_i) \\ \text{s.t.} &0 \le \delta_i \le 1 \\ &0 \le \theta_k \le 1 \end{aligned} \tag{20}$$

Here, *Loss* represents the loss function, whereas $\theta_k$ and $\delta_i$ are the rule weights and attribute weights mentioned above.

*Loss* in Eq. (19) is the well-known cross-entropy loss function. The advantage of the cross-entropy loss function lies in its sensitivity to mispredictions: compared with the mean squared error, it imposes a greater penalty on incorrect predictions, thereby driving the model to learn rapidly. In this study, the cross-entropy loss function is selected as the objective function of the model:

$$Loss = -\frac{1}{N} \sum_{i=1}^{N} \sum_{k=1}^{K} y_{i,k} \log P\,(\hat{y}_i = k) \tag{21}$$

Here, $y_{i,k}$ is a binary variable (0 or 1), which takes a value of 1 when the $i$th sample belongs to the $k$th class, $k$ represents the number of classes, $N$ represents the number of samples, and $P\,(\hat{y}_i = k)$ denotes the predicted probability of the $i$th sample for the $k$th class.

The model proposed in this study employs the P-CMA-ES method for parameter optimization. This is because, compared with other algorithms such as random search [28], gradient descent [29], and Bayesian optimization [30], P-CMA-ES can generate solutions within the feasible region, ensuring that the solutions strictly satisfy the equality constraints [31,32]. The steps are as follows:

Step 1: Parameter initialization:

The initial parameter set $P^0$ is defined as follows:

$$P^0 = \{\theta_1, \ldots, \theta_L, \beta_{1,1}, \ldots, \beta_{L,N}, \delta_1, \ldots, \delta_M, \} \tag{22}$$

Step 2: Sampling

Sampling operations are performed to generate a population:

$$P_i^{g+1} \sim w^g + \epsilon^g \mathbb{N}\,(0, C^g)\, i = 1, \ldots, \lambda \tag{23}$$

where $P_i^{g+1}$ represents the $i$th solution in the offspring of the $(g + 1)$th generation, $w$ denotes the mean of the population, $\epsilon$ is the step size, $\mathbb{N}$ denotes the normal distribution, and $C^g$ is the covariance matrix of the $g$th generation.

Step 3: Prediction

Projection operations are performed to satisfy the following constraints:

$$P_i^{g+1}(1 + n_e \times (j-1) : n_e \times j) = P_i^{g+1}(1 + n_e \times (j-1)$$
$$: n_e \times j) - A_e^T \times (A_e \times A_e^T)^{-1} \times P_i^{g+1}(1 + n_e \times (j-1)$$
$$: n_e \times j) \times A_e \tag{24}$$

The hyperplane can be expressed as $P_i^{g+1}(1 + n_e \times (j-1) : n_e \times j) = 1$, where $n_e$ represents the number of variables in the equality constraints of the solution $P_i^g$, $j = 1, \ldots, N+1$ denotes the number of equality constraints in the solution $P_i^g$ and $A_e = [1 \ldots 1]_{1 \times N}$ is the parameter vector.

Step 4: Selection

Selection operations are performed to update the population mean $w^{g+1} = \sum_{i=1}^{\tau} h_i P_{i:\lambda}^{g+1}$, where $P_{i:\lambda}^{g+1}$ represents the $i$th solution among $\lambda$ solutions in the $(g+1)$ th generation $\tau$ denotes the offspring population size.

Step 5: Adaptation operations

Adaptation operations are executed to update the covariance matrix.

   The above optimization process is recursively iterated until the optimal solution *Pbest* is obtained. The optimal EBRB model can then be constructed.

## 4. Experiment and analysis

This section is divided into three parts. In Part 1, the proposed method is applied to diabetes prediction. Part 2: The effectiveness of the method is validated through two experiments. The algorithm in this paper is implemented via the MATLAB programming language.

### 4.1. Case study

**4.1.1. Experiment definition.** To validate the effectiveness of the proposed NCR-EBRB diabetes diagnosis model, this section conducts an experimental analysis using real-world clinical diabetes diagnostic data. [33]. The data were collected from Iraqi society, as they were acquired from the laboratory of Medical City Hospital and (the Specializes Center for Endocrinology and Diabetes-Al-Kindy Teaching Hospital).

   The dataset comprises 1000 patients and covers three classes, namely, diabetic, nondiabetic, and predicted diabetic, with 844, 103, and 53 patients, respectively. Detailed information on the data parameters is presented in Table 1.

   In the experiments of this section, comparative experiments are conducted with decision trees, two ensemble models of decision trees (Bagging Tree and Boosting Tree), K-Nearest Neighbor (KNN), Support Vector Machine (SVM), and the single EBRB model to demonstrate the advancement of the proposed model.

**4.1.2. Model construction and parameter setting.** Equations (7)-(10) demonstrate the generation process of the EBRB rules. Among the subjectively set parameters are the attribute weights $\delta_i$, rule weights $\theta_k$, and utility values $u(A_{i,j})$. In the original EBRB, the attribute weights $\delta_i$ and rule weights $\theta_k$ are assigned by experts. In the experiments of this study, the original values are set to 1. For $u(A_{i,j})$, the utility values are generated through clustering from the data, with the quantity set to three.

   The modeling steps of the NCR-EBRB applied to diabetes prediction can be described as follows:

Step 1: Dataset partitioning

To demonstrate the effectiveness of the proposed model, the dataset is divided into training and testing sets using a 70%:30% split. The test set is extracted from each class of data to ensure balanced representation.

**Table 1. Overview of Patient Distribution and Data Parameters in the Study Dataset.**

| Pathological feature | Type |
|---|---|
| Gender | Nominal |
| AGE | Numeric |
| Urea | Numeric |
| Creatinine ratio(Cr) | Numeric |
| HbA1c | Numeric |
| Cholesterol (Chol) | Numeric |
| TG | Numeric |
| HDL | Numeric |
| LDL | Numeric |
| VLDL | Numeric |
| Body Mass Index (BMI) | Numeric |
| CLASS | Nominal |

## Step 2: Feature Selection

For the NCR-EBRB model, the XGBoost feature selection technique is employed to evaluate feature importance. The evaluation results are shown in Table 2, where the importance of the most critical feature is set to 1, and the importance scores of other features are derived relative to this baseline.

## Step 3: Setting Reference Values for the EBRB Model to Generate the Rule Base

Antecedent reference values $A_{i,j}$ are determined based on the selected features. Consequent reference values are determined by the classes. Since the dataset contains three classes, the reference values for the classes are set as follows: $\{m1, m2, m3\} = \{Diabetic, Non-Diabetic, Predicted-Diabetic\}$ the reference values of classes $mi$ are set as $\{0, 1, 2\}$.

## Step 4: Parameter initialization

In the original EBRB model, rule weights $\theta_k$, attribute weights $\delta_i$, and the belief distributions of rule consequents $\beta_n^k$ are assigned by experts.

In this study, the initial parameters $\theta_k$ and $\delta_i$ are set to 1. $\beta_n^k$ are determined by the sample classes. The hyperparameter $\lambda$ is set to 0.2.

**Table 2. Feature Importance Assessment Results.**

| feature | score |
|---|---|
| HbA1c | 1.00 |
| BMI | 0.68 |
| AGE | 0.22 |
| Chol | 0.18 |
| TG | 0.12 |
| VLDL | 0.04 |
| Cr | 0.04 |
| LDL | 0.02 |
| HDL | 0.01 |
| Urea | 0.01 |
| Gender | 0.01 |

Step 5: Rule reduction

After completing the configuration of the relevant parameters, the input data are processed to generate the original rule base. The NCR method proposed in Section 3.2 is subsequently applied to reduce the rule base, thereby generating a smaller and more refined new rule base.

Step 6: Parameter Optimization

The P-CMA-ES algorithm is utilized to optimize the parameters of the NCR-EBRB model established in Step 5. Finally, the optimized NCR-EBRB model is obtained.

The parameters of the NCR-EBRB model established in Step 5 are optimized using the P-CMA-ES. The algorithm employs a population size of $\lambda P-CMA-ES$ = 10 + 3 × ln$NP-CMA-ES$ (where $NP-CMA-ES$ is the number of decision variables), an initial step size of $\sigma P-CMA-ES$ = 0.5, and runs for a maximum of $GP-CMA-ES$ = 200 function evaluations as the termination criterion. The resulting optimized NCR-EBRB model is then used for inference.

For Step 2, the XGBoost feature analysis technique is employed for feature selection to construct a highly interpretable model. However, whether less important features should be included as input attributes for the model remains a question worth exploring. To address this, this study further investigates the impact of varying numbers of input attributes on model performance. Importantly, regardless of the number of input attributes, the selected features are always based on the importance ranking determined by the XGBoost feature analysis technique—i.e., the top features are sequentially selected according to their importance scores from highest to lowest. Table 2 presents the corresponding evaluation results, while Fig 4 illustrates the experimental outcomes for input attribute quantities of 2, 3, 4, and 5. Through comparative analysis, the relationship between the number of input attributes and model performance is more clearly revealed, thereby providing a more scientifically sound basis for feature selection.

As evidenced by the experimental results, the model exhibits optimal performance when the number of premise attributes is 2 or 4, whereas its effectiveness is relatively inferior when 3 or 5 attributes are used. This phenomenon suggests that, for complex data, the relationship between model performance and the number of input attributes is not a simple linear correlation but rather a more intricate nonlinear or composite relationship.

However, from the perspective of model interpretability, the NCR-EBRB, as an expert system model, inherently emphasizes its core strength in simulating the decision-making processes of human experts and presenting reasoning outcomes in a transparent and intuitive manner. When fewer input attributes are used, the system's decision logic becomes more concise and interpretable. This not only enhances user understanding and trust in the model but also reduces operational complexity in practical applications.

Therefore, in the context of applying the proposed NCR-EBRB model to diabetes diagnosis, a balanced consideration of experimental results and interpretability leads to the final selection of HbA1c (glycated hemoglobin) and BMI (body mass index) as the premise attributes. These two attributes are not only widely recognized in medical practice as critical indicators for diabetes diagnosis [34] but also capture the core characteristics of the disease in the most streamlined manner. This ensures robust model performance while maximizing transparency and practical utility.

**4.1.3. Experimental analysis.** After the NCR-EBRB model is established, the model parameters are altered due to rule reduction and parameter optimization. On the one hand, the number of rules is significantly reduced from 702 to 125. On the other hand, the rule weights and attribute weights are modified: the attribute weights are adjusted to 1 and 0.8893. The changes in rule weights are illustrated in Fig 5 (the original settings for both rule weights and attribute weights were 1).

To demonstrate the superiority of the proposed NCR-EBRB model, it is first compared with the original EBRB model. Owing to the presence of class imbalance in the dataset, a confusion matrix is employed to present the experimental results. The experimental results are shown in Fig 6.

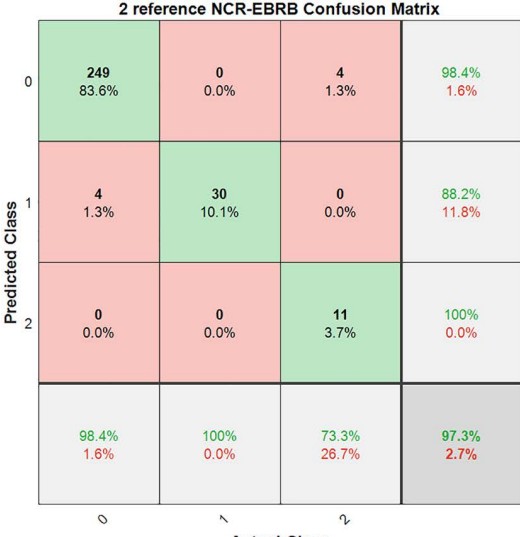
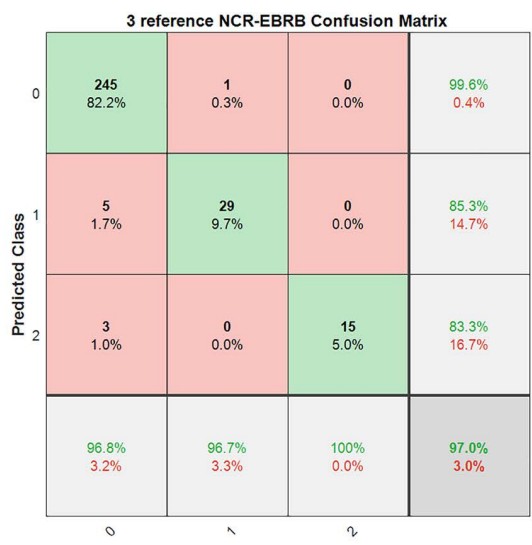
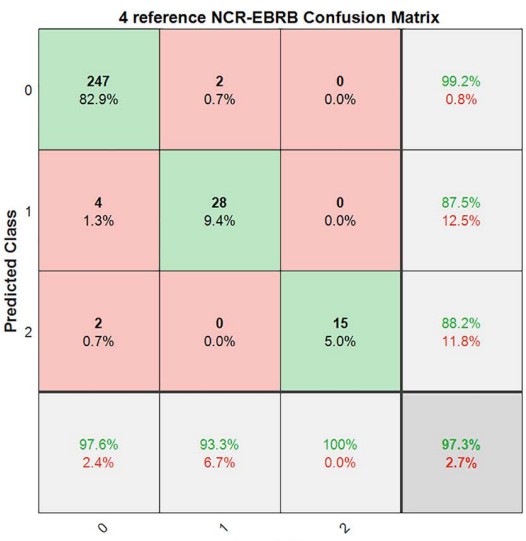
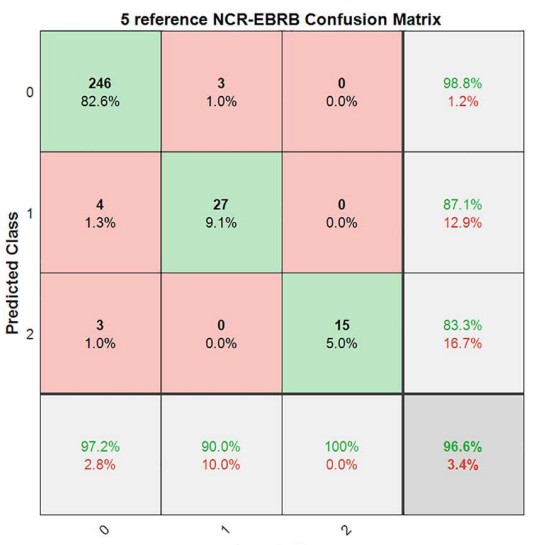

**Fig 4. Impact of the input attribute quantity on the performance of the NCR-EBRB model.**

The experimental results in Fig 6 show that the NCR-EBRB outperforms the EBRB model in predicting Class 0 and Class 2 while achieving performance identical to that of the EBRB model for Class 1. This is because the original data for Class 1 contain less noise, and the rule reduction process does not enhance model accuracy in this case.

To further verify the effectiveness of the proposed NCR-EBRB model, comparative experiments are conducted against five representative classifiers including Support Vector Machine (SVM), Decision Tree, K-Nearest Neighbor (KNN), Bagging Tree, Boosting Tree, and the single EBRB model. For reproducibility, all comparative models are implemented under fixed configurations without individual hyper-parameter tuning. Specifically, SVM adopts linear multi-class learning; Decision Tree is configured with moderate tree complexity; KNN employs a neighborhood size of five; Bagging Tree

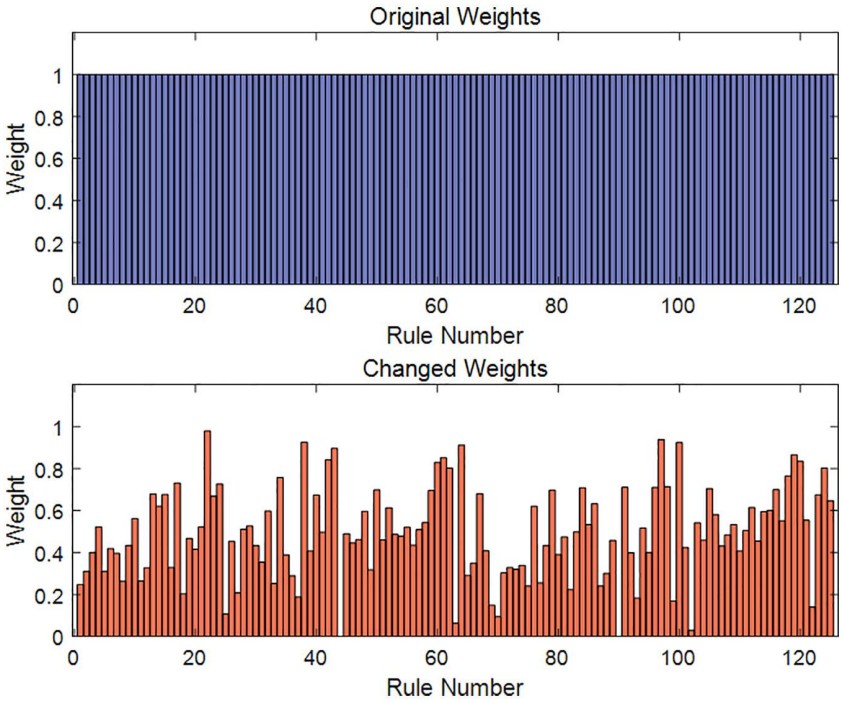

**Fig 5. Numerical distribution of rule weights optimized by the P-CMA-ES.**

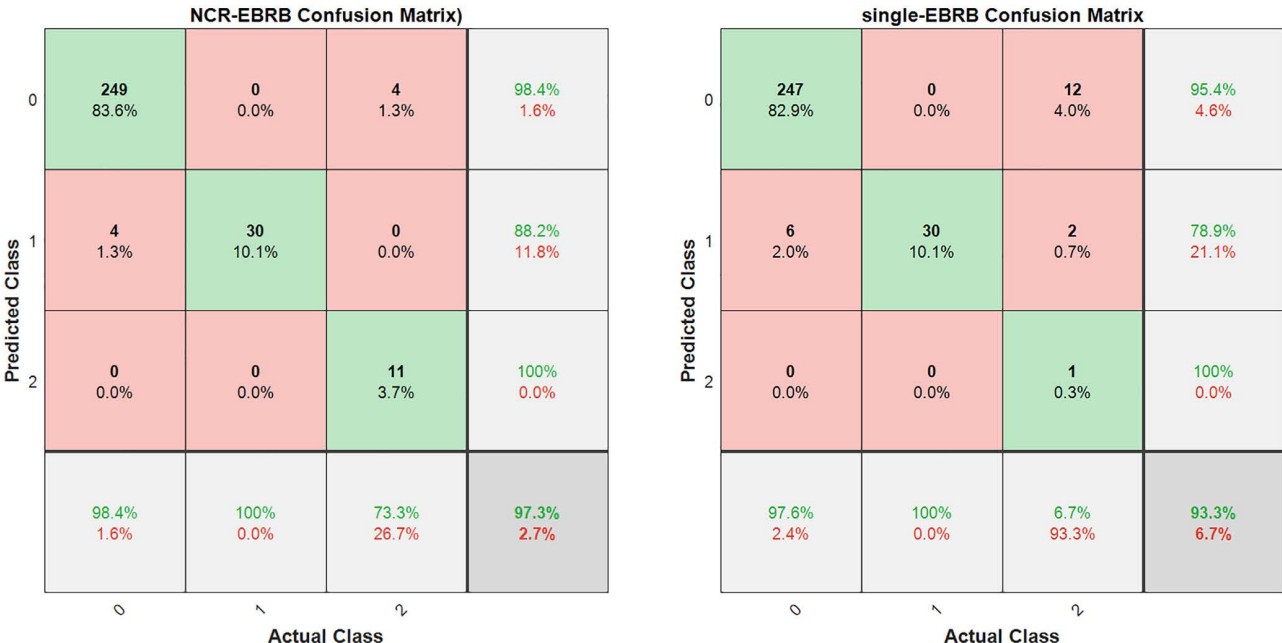

**Fig 6. Comparative performance analysis of the NCR-EBRB model and the original EBRB model.**

aggregates twenty base decision trees; and Boosting Tree is trained with decision trees as weak learners. Such configurations correspond to commonly used baseline settings and ensure a fair comparison framework.

Given the inherent class imbalance in the diabetes dataset, relying solely on globally aggregated true positives and false positives (i.e., micro-averaging) may mask the model's performance on minority classes. To provide a stringent and fair evaluation, the evaluation metrics in this study include overall accuracy, macro-averaged precision, and macro-averaged recall.

First, the overall accuracy is calculated to evaluate the holistic correctness across the entire dataset. Then, for precision and recall, we calculate the specific metrics for each individual class $i$ as follows:

$$Precision_i = \frac{TP_i}{TP_i + FP_i}, \quad Recall_i = \frac{TP_i}{TP_i + FN_i} \tag{25}$$

Where $TP_i, FP_i$ and $FN_i$ represent the true positives, false positives, and false negatives for the $i-$th class, respectively. The final macro-averaged precision and recall are then obtained by computing the arithmetic mean of these class-specific metrics across all $N$ classes:

$$Macro\_Precision = \frac{1}{N}\sum_{i=1}^{N} Precision_i$$
$$Macro\_recall = \frac{1}{N}\sum_{i=1}^{N} Recall_i \tag{26}$$

This macro-averaging strategy ensures that each diagnostic category (e.g., healthy vs. diabetic) is treated with equal weight, thereby providing a more rigorous assessment of the model's diagnostic robustness. The computed results are presented in Table 3.

From the experimental results, it is evident that the proposed NCR-EBRB model outperforms the original EBRB model across all the evaluation metrics, demonstrating its superior performance. Particularly in scenarios with limited test sample sizes, the NCR method is more accurate than other benchmark models are. However, as the scale of the test set expands and the volume of training data diminishes, the accuracy of the NCR-EBRB model gradually decreases. This phenomenon is attributed primarily to the application of the NCR rule reduction algorithm. While this algorithm effectively streamlines the rule base, it may also lead to an insufficient number of rules, thereby negatively impacting the model's overall accuracy.

**4.1.4. Ablation study.** To rigorously validate the individual contributions of the NCR module and P-CMA-ES optimization, a comprehensive ablation study was conducted across varying test set proportions (20%–40%), comparing three configurations: (1) Pure EBRB (no rule reduction/optimization), (2) NCR-EBRB without optimization (rule reduction only), and (3) the full NCR-EBRB (with both components).

As shown in Table 4, the NCR module alone improved the accuracy over the Pure EBRB at the 20% test split (0.9394 vs. 0.9343), reflecting effective rule pruning while preserving decision fidelity. Incorporating P-CMA-ES further enhanced performance significantly, increasing the accuracy to 0.9747 compared to 0.9394 for the unoptimized NCR-EBRB at the same split, indicating its necessity for fine-grained parameter adaptation.

Notably, the full NCR-EBRB model achieved its peak performance at the 25% split with an accuracy of 0.9759, outperforming the Pure EBRB (0.9197) by 5.62 percentage points. This confirms that both modules are indispensable for optimal predictive performance. Overall, the ablation results verify that NCR provides efficient structural reduction while P-CMA-ES unlocks additional optimization potential, and their combination yields robust performance across different partitioning ratios.

## 4.2. Benchmark

**4.2.1. Experimental definition.** To further validate the generalizability and adaptability of the NCR-EBRB method, four publicly available datasets are selected from the UCI Machine Learning Repository for experimental testing. Table 5

Table 3. Experimental Results under Different Testing Set Partitioning Strategies.

| Percentage | Model | Accuracy | Macro-Precision | Macro-Recall |
|---|---|---|---|---|
| 0.2 | NCR-EBRB | **0.9747** | **0.9229** | **0.9607** |
| | Bagging Tree | **0.9747** | **0.9071** | **0.9587** |
| | Boosting Tree | 0.9697 | 0.8960 | 0.9421 |
| | Decision Tree | 0.9545 | 0.8374 | 0.9194 |
| | SVM | 0.9343 | 0.7817 | 0.8488 |
| | KNN | 0.9596 | 0.8955 | 0.9254 |
| | EBRB | 0.9343 | 0.7810 | 0.8968 |
| 0.25 | NCR-EBRB | **0.9759** | **0.9268** | **0.9665** |
| | Bagging Tree | 0.9679 | 0.9375 | 0.9163 |
| | Boosting Tree | 0.9558 | 0.8976 | 0.8875 |
| | Decision Tree | 0.9398 | 0.8079 | 0.9164 |
| | SVM | 0.9237 | 0.7569 | 0.8139 |
| | KNN | 0.9398 | 0.8454 | 0.8817 |
| | EBRB | 0.9197 | 0.7411 | 0.8240 |
| | | | 0.7813 | |
| 0.3 | NCR-EBRB | **0.9732** | **0.9555** | **0.9058** |
| | Bagging Tree | 0.9631 | 0.9314 | 0.8947 |
| | Boosting Tree | 0.9597 | 0.9107 | 0.8934 |
| | Decision Tree | 0.9398 | 0.8228 | 0.9170 |
| | SVM | 0.9295 | 0.7667 | 0.8162 |
| | KNN | 0.9396 | 0.8278 | 0.8339 |
| | EBRB | 0.9329 | 0.7675 | 0.8273 |
| 0.35 | NCR-EBRB | **0.9628** | **0.8825** | **0.9169** |
| | Bagging Tree | 0.9570 | 0.9277 | 0.8740 |
| | Boosting Tree | 0.9542 | 0.9092 | 0.8728 |
| | Decision Tree | 0.9427 | 0.8191 | 0.8916 |
| | SVM | 0.9255 | 0.7649 | 0.7979 |
| | KNN | 0.9370 | 0.8377 | 0.8058 |
| | EBRB | 0.9255 | 0.7489 | 0.7712 |
| 0.4 | NCR-EBRB | 0.9549 | 0.9180 | 0.8733 |
| | Bagging Tree | **0.9599** | **0.9224** | **0.8973** |
| | Boosting Tree | 0.9574 | 0.9199 | 0.8814 |
| | Decision Tree | 0.9123 | 0.7572 | 0.8035 |
| | SVM | 0.9198 | 0.7601 | 0.7761 |
| | KNN | 0.9348 | 0.8290 | 0.8064 |
| | EBRB | 0.9248 | 0.7588 | 0.8072 |

details the fundamental information of the selected datasets, including dataset names, the number of input attributes, the number of classes, and the total sample size. In this section, the NCR-EBRB model is compared with several classical machine learning methods, specifically Decision Tree, Bagging Tree, Boosting Tree, k-Nearest Neighbor (k-NN), and Support Vector Machine (SVM). The experimental modeling workflow and parameter settings remain consistent with those in the previous section to ensure the fairness of the comparison.

**Table 4. Ablation Study of NCR-EBRB under Varying Test Set Partitioning Ratios.**

| Percentage | Model | Accuracy | Macro-Precision | Macro-Recall |
|---|---|---|---|---|
| 0.2 | NCR-EBRB | **0.9747** | **0.9229** | **0.9607** |
| | NCR-EBRB (without optimization) | 0.9394 | 0.8637 | 0.8087 |
| | Pure EBRB | 0.9343 | 0.7810 | 0.8968 |
| 0.25 | NCR-EBRB | **0.9759** | **0.9268** | **0.9665** |
| | NCR-EBRB (without optimization) | 0.9398 | 0.8454 | 0.8817 |
| | Pure EBRB | 0.9197 | 0.7411 | 0.8240 |
| 0.3 | NCR-EBRB | **0.9732** | **0.9555** | **0.9058** |
| | NCR-EBRB (without optimization) | 0.9329 | 0.8104 | 0.7895 |
| | Pure EBRB | 0.9329 | 0.7675 | 0.8273 |
| 0.35 | NCR-EBRB | **0.9628** | **0.8825** | **0.9169** |
| | NCR-EBRB (without optimization) | 0.9284 | 0.8153 | 0.7502 |
| | Pure EBRB | 0.9255 | 0.7489 | 0.7712 |
| 0.40 | NCR-EBRB | 0.9549 | 0.9180 | 0.8733 |
| | NCR-EBRB (without optimization) | 0.9348 | 0.8290 | 0.8064 |
| | Pure EBRB | 0.9248 | 0.7588 | 0.8072 |

**Table 5. Dataset overview.**

| Dataset | Number of Antecedent Attribute | Number of Classified Results | Sample Size |
|---|---|---|---|
| Iris | 4 | 3 | 150 |
| Banknote | 4 | 2 | 1372 |
| Ecoli | 7 | 2 | 336 |
| Newthyroid | 5 | 3 | 215 |

**4.2.2. Analysis results.** The experimental results are presented in Table 6. As shown in the table, the proposed NCR-EBRB model outperforms the other benchmark models across the four datasets (Newthyroid, Ecoli, Iris, and Banknote), demonstrating satisfactory classification performance. Therefore, in certain specific application scenarios, the NCR-EBRB can serve as an effective classification method to achieve desirable classification outcomes.

**4.2.3. Discussion of the value of the hyperparameter $\lambda$.** In the proposed NCR-EBRB system, a hyperparameter $\lambda$ is introduced to control the number of fused rules. To further investigate the impact of this parameter on model performance, the four datasets from Section 4.2 are selected to evaluate the effects of varying $\lambda$ values on system performance. In each experiment, the $\lambda$ parameter is adjusted as the independent variable, while all other parameters remain fixed to ensure experimental controllability and result comparability.

The experimental results are shown in Fig 7, where the horizontal axis represents the values of the hyperparameter $\lambda$, and the vertical axis indicates the classification accuracy of the NCR-EBRB system. The figure shows that for the Iris, Newthyroid, and Ecoli datasets, the system's classification accuracy tends to fluctuate as $\lambda$ varies, with multiple local maxima emerging during the process. Overall, the model achieves optimal classification performance when $\lambda = 0.2$ for the Banknote dataset; however, the accuracy curve displays only one distinct maximum, reaching peak accuracy at $\lambda = 0.9$.

This experiment further explores the impact of the $\lambda$ parameter on system performance, validating the complexity and necessity of its selection process. The results demonstrate that simply reducing $\lambda$ does not consistently improve model

**Table 6. Results of comparative research.**

| Dataset | Model | Accuracy | Macro-Precision | Macro-Recall |
|---|---|---|---|---|
| Iris | NCR-EBRB | **0.9778** | **0.9792** | **0.9778** |
| | BaggingTree | 0.9556 | 0.9608 | 0.9556 |
| | BoostingTree | 0.9333 | 0.9345 | 0.9333 |
| | DecisionTree | 0.8889 | 0.8899 | 0.8889 |
| | SVM | 0.9556 | 0.9608 | 0.9556 |
| | KNN | 0.9556 | 0.9608 | 0.9556 |
| Banknote | NCR-EBRB | **0.9538** | **0.9524** | **0.9546** |
| | BaggingTree | 0.9367 | 0.9353 | 0.9371 |
| | BoostingTree | 0.9416 | 0.9405 | 0.9414 |
| | DecisionTree | 0.9392 | 0.9386 | 0.9382 |
| | SVM | 0.8662 | 0.8664 | 0.8621 |
| | KNN | 0.9465 | 0.9454 | 0.9464 |
| Ecoli | NCR-EBRB | **0.9600** | **0.8571** | **0.9778** |
| | BaggingTree | 0.9300 | 0.8114 | 0.7833 |
| | BoostingTree | 0.9000 | 0.7273 | 0.7667 |
| | DecisionTree | 0.9400 | 0.8533 | 0.7889 |
| | SVM | 0.9000 | 0.7381 | 0.8556 |
| | KNN | 0.9200 | 0.7778 | 0.7778 |
| Newthyroid | NCR-EBRB | **0.9688** | **0.9444** | **0.9556** |
| | BaggingTree | 0.9531 | 0.9438 | 0.9185 |
| | BoostingTree | 0.8906 | 0.8662 | 0.8667 |
| | DecisionTree | 0.8438 | 0.7736 | 0.8444 |
| | SVM | 0.9375 | 0.9296 | 0.9111 |
| | KNN | 0.9063 | 0.8944 | 0.8704 |

accuracy. This may occur because excessively small $\lambda$ values filter out critical information that is beneficial for classification. Similarly, blindly increasing $\lambda$ does not guarantee better performance, as it may introduce excessive noise that interferes with decision-making. Additionally, as $\lambda$ increases, the model's computational time increases accordingly. Therefore, in practical applications, the value of $\lambda$ should be carefully selected on the basis of specific data characteristics to achieve an optimal balance between accuracy and efficiency.

**4.2.4. Experimental summary.** This section validates the generalizability and effectiveness of the proposed NCR-EBRB model via four public datasets (Iris, Banknote, Ecoli, and Newthyroid). The experimental results demonstrate that the NCR-EBRB significantly outperforms benchmark models in terms of accuracy, precision, and recall across all datasets, further confirming its superior performance in classification tasks.

Furthermore, the analysis of the impact of the hyperparameter $\lambda$ on model performance reveals significant variations in the optimal $\lambda$ values across different datasets. For example, Iris, Ecoli, and Newthyroid achieve peak performance at $\lambda = 0.2$ values across different datasets. Banknote achieve peak performance at $\lambda = 0.9$. This phenomenon highlights the sensitivity of the NCR-EBRB model to the rule fusion threshold: excessively low $\lambda$ values may filter out critical information, whereas excessively high values may introduce redundant noise, degrading classification decisions. Therefore, in practical applications, $\lambda$ should be carefully configured on the basis of specific data characteristics. Future research could explore adaptive adjustment mechanisms to enhance the model's generalization ability across diverse scenarios.

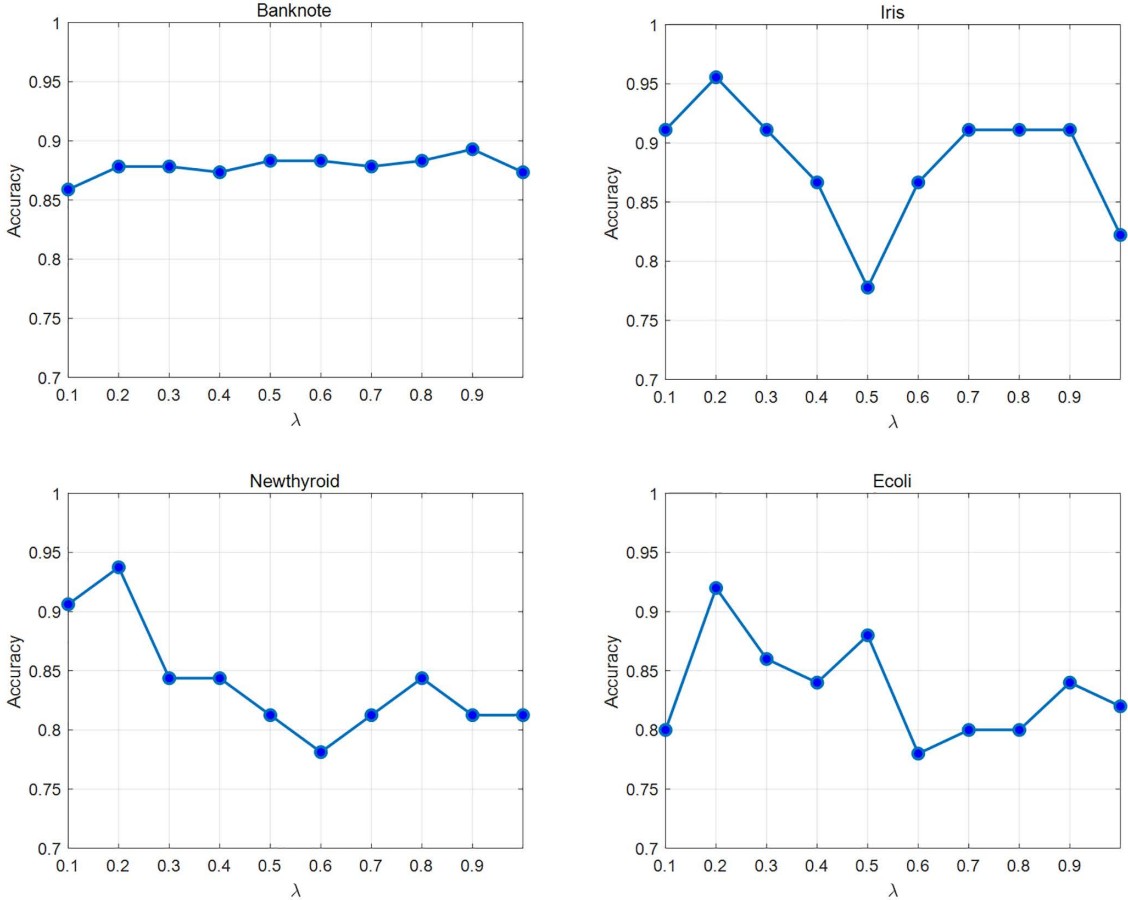

**Fig 7. Effects of the parameter $\lambda$ on the experimental performance.**

In summary, the NCR-EBRB method exhibits excellent classification performance and robustness in most experimental settings, effectively validating the feasibility and practicality of the neighborhood coverage reduction strategy and parameter optimization method.

## 5. Conclusion

This study proposes a feature selection-based neighborhood coverage reduction EBRB (NCR-EBRB) method, which simultaneously addresses critical challenges in feature definition and initial parameter configuration. Experimental validation across four benchmark datasets demonstrates its robust performance and strong application potential in classification tasks.

The model's performance is inherently sensitive to the hyperparameter $\lambda$ (rule fusion threshold), as $\lambda$ directly governs the final rule base size (Section 4.2.3). An ill-chosen $\lambda$ may discard critical diagnostic rules or retain redundant ones, potentially degrading accuracy.

The core innovation of NCR-EBRB—rule base simplification—significantly enhances interpretability without sacrificing performance. By reducing the rule count from 702 to 125 (an 82.2% reduction), the model achieves a highly transparent and clinically actionable rule set while maintaining high accuracy. This drastic simplification directly fulfills the primary

objective of improving model interpretability, enabling medical experts to trace and validate diagnostic reasoning with confidence.

To further strengthen stability, future work will develop intelligent parameter adjustment mechanisms to reduce manual tuning reliance. We also plan to integrate local interpretability techniques, [35] for deeper analysis of the NCR-EBRB decision process, enhancing its clinical utility and transparency.

## Supporting information

**S1 File. Supporting information datasets (Supporting Information.zip).** This ZIP file contains all experimental datasets used in this study, organized as follows: 1. Original Data: Sourced from the Medical City Hospital/Al-Kindy Teaching Hospital (Mendeley Data) and the UCI Machine Learning Repository. 2. Section 4.1 Datasets: Training and testing splits for diabetes diagnosis with test set proportions of 0.2, 0.25, 0.3, 0.35, and 0.4 (e.g., Dataset of Diabetes_train_0.2.xlsx). 3. Section 4.2 Datasets: Validation datasets including Iris, Banknote, Ecoli, and Newthyroid with a 0.3 test split. All processed datasets are restricted to two premise attributes to ensure a fair comparison by eliminating the influence of feature selection, thereby highlighting the performance of the NCR-EBRB model.
(ZIP)

## Author contributions

**Conceptualization:** Shucheng Feng.

**Data curation:** Li Jiang, Manlin Chen.

**Formal analysis:** Shucheng Feng, Manlin Chen.

**Funding acquisition:** Wei He.

**Investigation:** Li Jiang, Manlin Chen.

**Methodology:** Shucheng Feng.

**Project administration:** Manlin Chen.

**Resources:** Li Jiang, Manlin Chen.

**Supervision:** Manlin Chen.

**Validation:** Shucheng Feng, Wei He.

**Visualization:** Manlin Chen.

**Writing – original draft:** Shucheng Feng.

**Writing – review & editing:** Wei He.

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
