## [Decision Letter · Decision Letter 0]

12 Jan 2026

PONE-D-25-53239A new extended belief rule base method based on neighborhood covering reduction for diabetes diagnosisPLOS One

Dear Dr. He,

Thank you for submitting your manuscript to PLOS ONE. After careful consideration, we feel that it has merit but does not fully meet PLOS ONE’s publication criteria as it currently stands. Therefore, we invite you to submit a revised version of the manuscript that addresses the points raised during the review process.

If applicable, we recommend that you deposit your laboratory protocols in protocols.io to enhance the reproducibility of your results. Protocols.io assigns your protocol its own identifier (DOI) so that it can be cited independently in the future. For instructions see: https://journals.plos.org/plosone/s/submission-guidelines#loc-laboratory-protocols. Additionally, PLOS ONE offers an option for publishing peer-reviewed Lab Protocol articles, which describe protocols hosted on protocols.io. Read more information on sharing protocols at . Additionally, PLOS ONE offers an option for publishing peer-reviewed Lab Protocol articles, which describe protocols hosted on protocols.io. Read more information on sharing protocols at https://plos.org/protocols?utm_medium=editorial-email&utm_source=authorletters&utm_campaign=protocols..

We look forward to receiving your revised manuscript.

Kind regards,

Robin Haunschild

Academic Editor

PLOS One

Journal Requirements:

When submitting your revision, we need you to address these additional requirements. 1. Please ensure that your manuscript meets PLOS ONE's style requirements, including those for file naming. The PLOS ONE style templates can be found at  https://journals.plos.org/plosone/s/file?id=wjVg/PLOSOne_formatting_sample_main_body.pdf and https://journals.plos.org/plosone/s/file?id=ba62/PLOSOne_formatting_sample_title_authors_affiliations.pdf 2. We suggest you thoroughly copyedit your manuscript for language usage, spelling, and grammar. If you do not know anyone who can help you do this, you may wish to consider employing a professional scientific editing service.  The American Journal Experts (AJE) (https://www.aje.com/) is one such service that has extensive experience helping authors meet PLOS guidelines and can provide language editing, translation, manuscript formatting, and figure formatting to ensure your manuscript meets our submission guidelines. Please note that having the manuscript copyedited by AJE or any other editing services does not guarantee selection for peer review or acceptance for publication.  Upon resubmission, please provide the following: • The name of the colleague or the details of the professional service that edited your manuscript• A copy of your manuscript showing your changes by either highlighting them or using track changes (uploaded as a *supporting information* file)• A clean copy of the edited manuscript (uploaded as the new *manuscript* file) 3. Please note that PLOS One has specific guidelines on code sharing for submissions in which author-generated code underpins the findings in the manuscript. In these cases, we expect all author-generated code to be made available without restrictions upon publication of the work.  Please review our guidelines at https://journals.plos.org/plosone/s/materials-and-software-sharing#loc-sharing-code and ensure that your code is shared in a way that follows best practice and facilitates reproducibility and reuse. 4. Thank you for stating in your Funding Statement:  “This work was supported in part by Open Foundation of Key Laboratory of the Ministry of Education on Application of Artificial Intelligence in Equipment under Grant No.AAIE-2023-0103，in part by the Natural Science Foundation of Heilongjiang Province under Grant No. PL2024G009, in part by the Basic Research Support Program for Outstanding Young Teachers in Provincial Undergraduate Universities of Heilongjiang Province under Grant No. YQJH2024116.” Please provide an amended statement that declares *all* the funding or sources of support (whether external or internal to your organization) received during this study, as detailed online in our guide for authors at http://journals.plos.org/plosone/s/submit-now. Please also include the statement “There was no additional external funding received for this study.” in your updated Funding Statement.  Please include your amended Funding Statement within your cover letter. We will change the online submission form on your behalf. 5. Thank you for stating the following financial disclosure:  “This work was supported in part by Open Foundation of Key Laboratory of the Ministry of Education on Application of Artificial Intelligence in Equipment under Grant No.AAIE-2023-0103，in part by the Natural Science Foundation of Heilongjiang Province under Grant No. PL2024G009, in part by the Basic Research Support Program for Outstanding Young Teachers in Provincial Undergraduate Universities of Heilongjiang Province under Grant No. YQJH2024116.” Please state what role the funders took in the study.  If the funders had no role, please state: "The funders had no role in study design, data collection and analysis, decision to publish, or preparation of the manuscript." If this statement is not correct you must amend it as needed.  Please include this amended Role of Funder statement in your cover letter; we will change the online submission form on your behalf. 6. Please note that funding information should not appear in the Acknowledgments section or other areas of your manuscript. We will only publish funding information present in the Funding Statement section of the online submission form. Please remove any funding-related text from the manuscript. 

Reviewers' comments:

Reviewer's Responses to Questions

**Comments to the Author**

1. Is the manuscript technically sound, and do the data support the conclusions?

Reviewer #1: Yes

Reviewer #2: Yes

Reviewer #3: Yes

2. Has the statistical analysis been performed appropriately and rigorously? 

Reviewer #1: Yes

Reviewer #2: Yes

Reviewer #3: Yes

3. Have the authors made all data underlying the findings in their manuscript fully available?

Reviewer #1: Yes

Reviewer #2: Yes

Reviewer #3: Yes

4. Is the manuscript presented in an intelligible fashion and written in standard English?

Reviewer #1: Yes

Reviewer #2: Yes

Reviewer #3: Yes

5. Review Comments to the Author

Reviewer #1: The following points should be addressed to improve the clarity, rigor, and overall impact of the manuscript:

1.Data Description Inaccuracy (Section 4.1):

Comment: The text states, "The dataset comprises 1000 patients... with 844, 103, and 5 patients, respectively." This sums to 952, not 1000. Please verify and correct the total number of samples in the dataset to ensure accuracy.

2.Formula Numbering:

Comment: There is a duplication in equation numbering; two distinct equations are both labeled as "(7)". Please renumber all equations sequentially throughout the manuscript to avoid confusion.

3.Terminology Consistency:

Comment: The model name alternates between the full form "Extended Belief Rule Base" and the acronym "EBRB". For consistency and readability, it is recommended to use the acronym "EBRB" throughout the manuscript after it is first defined in the abstract or introduction.

4.Lack of Optimization Details (P-CMA-ES):

Comment: The description of the P-CMA-ES optimization process lacks critical implementation details, such as the termination criteria and population size. To enhance reproducibility, please specify all key hyperparameters used. Furthermore, providing a convergence curve of the optimization algorithm would help demonstrate that a stable solution was reached.

5.Need for Ablation Study:

Comment: The individual contributions of the NCR module and the P-CMA-ES optimization to the model's final performance are not evaluated. To disentangle their effects, it is strongly recommended to incorporate an ablation study. This should compare the performance of:

The pure EBRB model (without rule reduction).

The NCR-EBRB model (with rule reduction but without parameter optimization).

The full NCR-EBRB model (with both rule reduction and optimization).

This analysis is crucial for validating the necessity of each proposed component.

6.Missing Limitations Analysis:

Comment: The conclusion section would be strengthened by a discussion of the model's limitations. For instance, the performance appears sensitive to hyperparameters (e.g., the rule fusion threshold t). Furthermore, as hinted at in Table 3 with the 40% test set, rule reduction may lead to performance degradation when training data is scarce. Acknowledging these limitations provides a more balanced and scientifically honest perspective.

7.Strengthening the Conclusion:

Comment: The conclusion should more directly address the research objectives stated in the introduction. If "improving interpretability" was a key goal, the discussion should explicitly state how NCR-EBRB achieves this. For example, quantify the improvement by stating that the model "reduced the number of rules by 50% while maintaining 95% accuracy," thereby enhancing transparency and traceability.

Reviewer #2: 1. Formal theoretical justification for integrating NCR with EBRB is lacking

The manuscript introduces Neighborhood Covering Reduction (NCR) for EBRB rule reduction, yet the presentation remains largely algorithmic. A theoretical analysis of how this reduction affects the reasoning consistency or completeness of the EBRB system is absent. It is recommended that the authors include a discussion on topics such as rule coverage sufficiency, information preservation, or potential error bounds to strengthen the theoretical foundation.

2. Semantic consistency after rule reduction is not clearly addressed

In EBRB systems, rules encapsulate expert semantic knowledge, while NCR operates as a distance based rule selection method. The manuscript does not examine whether rule removal via NCR preserves the original semantic integrity of the expert knowledge. A discussion on the semantic preservation characteristics of the proposed NCR EBRB framework would therefore be valuable.

3. No automatic selection mechanism for the rule fusion threshold λ

Although Section 4.2 analyzes the influence of the rule fusion threshold λ, its determination in practice still depends on manual tuning. It is suggested that the authors consider validation based or adaptive strategies to automatically select an appropriate λ, thereby improving the usability and robustness of the method.

4. Worst case computational complexity of the rule reduction algorithm is not analyzed

Algorithm 1 employs a greedy forward search strategy; however, its worst case time and space complexity for large scale rule bases is not discussed. Providing a formal complexity analysis would help clarify the scalability of the proposed approach.

5. Identical Precision and Recall values require explanation

In Tables 3 and 5, Precision and Recall are repeatedly reported as equal. Such a result is uncommon in general classification tasks and warrants clarification—for instance, whether it stems from macro averaging, class imbalance, or specific implementation details.

6. Parameter settings of comparative models are not disclosed

The parameter configurations for the comparative models (SVM, KNN, Bagging Tree, and Boosting Tree) are not provided. This omission may compromise the fairness and reproducibility of the experimental comparisons. Reporting these details is essential for rigorous evaluation.

7. Section lengths are imbalanced

Section 3.2 is relatively lengthy compared to the concise Conclusion section. Rebalancing the manuscript structure would enhance overall readability and coherence.

8. Inconsistent figure/table numbering and textual references

Some figures and tables are inconsistently numbered or referenced in the main text. A thorough check to ensure consistency is recommended.

Reviewer #3: This manuscript proposes an extended belief rule base (EBRB) framework for diabetes diagnosis. The topic is relevant to clinical decision support, and the integration of rule-base reduction with an interpretable EBRB-style model is potentially useful. The paper is suitable for publication after minor revisions that address the following issues.

1. The manuscript states that NCR-EBRB outperforms other benchmark models across the four UCI datasets. However, Table 5 indicates that for Banknote, KNN (0.9075) exceeds NCR-EBRB (0.8929).

2. Equation (7) is used for the XGBoost gain function and later reused for the EBRB rule expression.

3. Algorithm 1 repeats step number “7” and has minor formatting issues.

4. Several visible typographical and spacing issues appear.

6. PLOS authors have the option to publish the peer review history of their article (what does this mean?). If published, this will include your full peer review and any attached files.). If published, this will include your full peer review and any attached files.

.

Reviewer #1: No

Reviewer #2: No

Reviewer #3: No

---

## [Author Response · Author response to Decision Letter 1]

11 Feb 2026

Due to formatting constraints, our detailed responses to the specific comments from the reviewers and the academic editor are provided in the separate "Response to Reviewers" file.

---

## [Editor Report · Decision Letter 1]

15 Feb 2026

PONE-D-25-53239R1A new extended belief rule base method based on neighborhood covering reduction for diabetes diagnosisPLOS One

Dear Dr. He,

Thank you for submitting your manuscript to PLOS ONE. After careful consideration, we feel that it has merit but does not fully meet PLOS ONE’s publication criteria as it currently stands. Therefore, we invite you to submit a revised version of the manuscript that addresses the points raised during the review process.

If applicable, we recommend that you deposit your laboratory protocols in protocols.io to enhance the reproducibility of your results. Protocols.io assigns your protocol its own identifier (DOI) so that it can be cited independently in the future. For instructions see: https://journals.plos.org/plosone/s/submission-guidelines#loc-laboratory-protocols. Additionally, PLOS ONE offers an option for publishing peer-reviewed Lab Protocol articles, which describe protocols hosted on protocols.io. Read more information on sharing protocols at . Additionally, PLOS ONE offers an option for publishing peer-reviewed Lab Protocol articles, which describe protocols hosted on protocols.io. Read more information on sharing protocols at https://plos.org/protocols?utm_medium=editorial-email&utm_source=authorletters&utm_campaign=protocols..

We look forward to receiving your revised manuscript.

Kind regards,

Robin Haunschild

Academic Editor

PLOS One

Journal Requirements:

Additional Editor Comments

Please update your submission files as you proposed.

---

## [Author Response · Author response to Decision Letter 2]

20 Feb 2026

Please refer to the uploaded file "Response to Reviewers" for our point-by-point responses to the academic editor and reviewers' comments.

---

## [Decision Letter · Decision Letter 2]

31 Mar 2026

A new extended belief rule base method based on neighborhood covering reduction for diabetes diagnosis

PONE-D-25-53239R2

Dear Dr. He,

We’re pleased to inform you that your manuscript has been judged scientifically suitable for publication and will be formally accepted for publication once it meets all outstanding technical requirements.

An invoice will be generated when your article is formally accepted. Please note, if your institution has a publishing partnership with PLOS and your article meets the relevant criteria, all or part of your publication costs will be covered. Please make sure your user information is up-to-date by logging into Editorial Manager at Editorial Manager® and clicking the ‘Update My Information' link at the top of the page. For questions related to billing, please contact  and clicking the ‘Update My Information' link at the top of the page. For questions related to billing, please contact billing support..

Kind regards,

Robin Haunschild

Academic Editor

PLOS One

Additional Editor Comments (optional):

Reviewers' comments:

Reviewer's Responses to Questions

**Comments to the Author**

1. If the authors have adequately addressed your comments raised in a previous round of review and you feel that this manuscript is now acceptable for publication, you may indicate that here to bypass the “Comments to the Author” section, enter your conflict of interest statement in the “Confidential to Editor” section, and submit your "Accept" recommendation.

Reviewer #1: All comments have been addressed

Reviewer #2: All comments have been addressed

Reviewer #3: All comments have been addressed

2. Is the manuscript technically sound, and do the data support the conclusions?

Reviewer #1: Yes

Reviewer #2: Yes

Reviewer #3: Yes

3. Has the statistical analysis been performed appropriately and rigorously? 

Reviewer #1: Yes

Reviewer #2: Yes

Reviewer #3: N/A

4. Have the authors made all data underlying the findings in their manuscript fully available?

Reviewer #1: Yes

Reviewer #2: Yes

Reviewer #3: Yes

5. Is the manuscript presented in an intelligible fashion and written in standard English?

Reviewer #1: Yes

Reviewer #2: Yes

Reviewer #3: Yes

6. Review Comments to the Author

Reviewer #1: All comments have been revised, and the manuscript can be accepted. This paper has certain innovation and application prospects.

Reviewer #2: The authors have revised the manuscript according to the comments, and the quality has been improved significantly.

Reviewer #3: The authors have addressed all my comments. I have no further questions. I would recommend accepting this paper.

7. PLOS authors have the option to publish the peer review history of their article (what does this mean?). If published, this will include your full peer review and any attached files.). If published, this will include your full peer review and any attached files.

.

Reviewer #1: No

Reviewer #2: No

Reviewer #3: No

---

## [Editor Report · Acceptance letter]

PONE-D-25-53239R2

PLOS One

Dear Dr. He,

I'm pleased to inform you that your manuscript has been deemed suitable for publication in PLOS One. Congratulations! Your manuscript is now being handed over to our production team.

Kind regards,

on behalf of

Dr. Robin Haunschild

Academic Editor

PLOS One